# Residential Segregation and Epigenetic Age Acceleration Among Older-Age Black and White Americans

**DOI:** 10.3390/ijerph22060837

**Published:** 2025-05-27

**Authors:** Reed DeAngelis, Victoria Fisher, John Dou, Kelly Bakulski, David Rigby, Margaret Hicken

**Affiliations:** 1Institute for Social Research, University of Michigan, Ann Arbor, MI 48104, USA; rigby@umich.edu (D.R.); mhicken@umich.edu (M.H.); 2College of Human Medicine, Michigan State University, East Lansing, MI 48824, USA; fisherv1@msu.edu; 3Department of Epidemiology, University of Michigan, Ann Arbor, MI 48109, USA; johndou@umich.edu (J.D.); bakulski@umich.edu (K.B.)

**Keywords:** aging, DNA methylation, epigenetic age, GrimAge, racial disparities, residential segregation

## Abstract

Our study tests residential segregation as an explanation for biological aging disparities between Black and White Americans. We analyze data from 288 Black and White older-age adults who participated in Wave 6 (2019) of the Americans’ Changing Lives study, a nationally representative cohort of adults in the contiguous United States. Our outcome of interest is epigenetic age acceleration assessed via five epigenetic clocks: GrimAge, PhenoAge, SkinBloodAge, HannumAge, and HorvathAge. Residential segregation is operationalized at the census tract level using the Getis-Ord G_i_* statistic and multilevel modeling procedures that adjust for state-level clustering. We uncover three key findings. First, epigenetic age profiles are comparable among White respondents regardless of where they live. Second, Black respondents express roughly three years of accelerated epigenetic age (GrimAge), relative to White counterparts, regardless of where they live. Third, diminished education levels and homeownership rates, coupled with elevated levels of traumatic stress and smoking, explain why Black residents in segregated Black areas exhibit accelerated epigenetic age. However, these factors do not explain why Black respondents living outside segregated Black areas also exhibit epigenetic age acceleration. Our findings suggest residential segregation only partially explains why Black Americans tend to live shorter lives than White Americans.

## 1. Introduction

Despite progress over the twentieth century, wide disparities in health and longevity persist among Black and White Americans. In 2019, Black Americans could expect to live 75 years at birth, compared with 79 years for their White peers [1]. Black Americans can also expect to live more years with chronic aging-related conditions, including cardiovascular disease, kidney disease, diabetes, and cancer [2,3].

Black–White health disparities are often attributed to legacies of residential segregation in the United States (US) [4,5,6]. Indeed, decades of institutionalized housing discrimination (e.g., redlining), market-oriented policy reforms, and accompanying trends of White flight have resulted in starkly segregated residential contexts nationwide [7]. Access to higher education, high-paying jobs, and safe housing, in turn, has been concentrated disproportionately within affluent and majority-White communities [8,9]. Moreover, socioeconomic resources like education, money, and stable housing often determine access to quality healthcare and nutrition, as well as exposure to health hazards like crime and pollution [4,5]. Given that predominantly White communities tend to have fewer environmental risks and more health resources than their Black peers, it stands to reason that residential segregation can account for Black–White health disparities [4].

The aim of this study is to investigate whether and to what extent residential segregation accounts for Black–White disparities in biological aging. Assessed through DNA methylation cytosine–guanine dinucleotide (CpG) sites, biological age—hereafter, “epigenetic age”—is an aging biomarker reflecting the functional age of cells and tissues within the body, irrespective of chronological age [10]. Accelerated aging is a common factor in early mortality and chronic diseases known to plague Black communities [10]. Chronic exposure to socioeconomic disadvantage and psychosocial stress has also been linked to accelerated epigenetic aging [11], especially among Black adults [12,13]. Identifying social determinants of epigenetic aging is therefore a critical first step toward intervening on wider Black–White health disparities in the United States [14].

Our study advances the above aims by being one of few to test residential segregation as a social determinant of epigenetic aging disparities among a national sample of older-age Black and White adults [13,15]. Indeed, a scoping review found only nine studies conducted on the topic of neighborhood determinants of epigenetic age since early 2022 [16]. However, none of these studies focused on residential segregation. Following the 2022 review, one recent study found that older-age Black residents who lived in segregated and impoverished Black neighborhoods exhibited signs of accelerated epigenetic age, relative to their Black peers in more advantaged neighborhoods [17]. However, this study only included six US cities, whereas our analysis represents the entire contiguous US.

Drawing from prior studies, we also account for various individual characteristics that may explain the association between residential segregation and aging. These include educational attainment [9,18], homeownership [19], smoking [20], and stressful life events [21,22]. Though descriptive, our investigation provides novel findings that point to several avenues for future research and intervention.

## 2. Materials and Methods

### 2.1. Data

Data originate from Wave 6 (2019) of the Americans’ Changing Lives (ACL) cohort, a closed longitudinal study following a nationally representative sample of adults in the contiguous US [23]. Wave 6 of ACL was approved by the University of Michigan’s Institutional Review Board (#HUM00153243). The first wave of data collection (1986) recruited adults aged 25 and older through a multistage stratified area probability sample, with Black adults and people over the age of 60 oversampled at twice the rate of others [24]. To date, ACL has collected six waves of data at varying intervals. Survey data were collected from participants through extensive in-person interviews, covering numerous demographic and behavioral characteristics. Beginning in Wave 6, opt-in blood draws were collected to extract DNA methylation data [23]. Respondents’ geocoded addresses have also been recorded and linked with US census boundaries at each wave. Our analytic sample includes 288 non-Hispanic Black and White adults, aged 58 to 96 (mean = 71), who participated in Waves 1 and 6 and provided valid survey, biomarker, and residential data. Among this group, there were no missing observations on selected measures. All respondents provided written informed consent before participating in Waves 1 and 6.

### 2.2. Measures

*Respondent Race–Ethnicity*. Respondents were asked at Wave 1 to report (1) their race (White, Black, American Indian, Asian, or other), and (2) whether they were of Spanish or Hispanic descent. From these separate questions, we created two categories for self-identified non-Hispanic Black and White. Other racial–ethnic groups were excluded from our analyses due to small cell sizes and because our focus is on Black–White disparities.

*Racial Residential Segregation*. We measured residential segregation at the census tract level using the Getis-Ord i* (Gi*) hotspot statistic. The Gi* statistic returns a Z-score for each tract, reflecting the extent to which the racial–ethnic composition of the tract deviates from the average composition of the broader census-based statistical area (CBSA) or county in which the tract is located [25]. The Gi* statistics used in our analyses reflect queen contiguity scores that weight the composition of the focal tract by all neighboring tracts sharing a common vertex (i.e., edge or corner). Gi* Z-scores at or above 1.96 indicate significantly higher racial–ethnic clustering within and immediately surrounding the tract, relative to the larger CBSA or county [26].

Combining Gi* Z-scores with respondents’ race–ethnicity, we grouped Black and White respondents into discrete neighborhood categories denoting levels of Black or White residential clustering, respectively, within each respondent’s tract [27,28]. This included two groups for both Black and White respondents: (1) “highly clustered” (Z > 1.96); and (2) “non-clustered” (Z ≤ 1.96). When discussing our results, we refer to highly clustered and non-clustered tracts as “segregated” and “non-segregated”, respectively. We also report on sensitivity analyses that applied a more conservative segregation threshold of Z > 2.58. Researchers sometimes include a third “underrepresented” category denoting respondents who live in tracts where their racial–ethnic group is a minority (Z < −1.96) [27]. We did not include this category because no Black and only 16 White respondents lived in such tracts.

*Epigenetic Age Acceleration.* DNA methylation was quantified using the Illumina Infinium HumanMethylation EPIC BeadArray (version 1) [29]. Preprocessing of DNA methylation data samples was conducted using the “ewastools” package. To evaluate probe detection, background fluorescence was estimated from non-specific binding at completely methylated and unmethylated CpG sites [30]. At a threshold of *p* > 0.05 for detection-p, a total of 10,394 probes failed detection in >1% of samples and were removed. An additional 42,792 probes were removed for being previously flagged as cross-reactive [31]. After removal of probes failing detection, one sample had >1% of probes fail detection and was filtered. Remaining samples all had high intensity values (i.e., sum of median log2 intensity methylated and unmethylated signal > 20), and no mismatches with methylation-estimated sex. We evaluated relatedness using SNP probes on the methylation array, and no genetic matching of samples was observed. DNA methylation was noob background and dye-bias corrected [32], obtaining the final methylation data matrix of 315 samples and 813,099 CpG sites.

Five epigenetic age clocks were then calculated from participants’ DNA methylation data: GrimAge [33], Pheno/Levine Age [34], HannumAge [35], HorvathAge [36], and SkinBloodAge [37]. Cell composition was estimated using the “FlowSorted.Blood.EPIC” reference panel for deconvolution [38]. Estimated cell types included B cells, monocytes, neutrophils, natural killer cells, CD4 T lymphocytes, and CD8 T lymphocytes. Epigenetic age acceleration reflects the residuals from regressing each epigenetic clock on participants’ chronological age at the time of their Wave 6 interview. Positive and negative residuals indicate accelerated and decelerated aging, respectively, while values of zero represent equal chronological and epigenetic age.

*Covariates.* All models included covariates for *respondent sex* (female vs. male) and *cell composition* from the Wave 6 blood draw (% monocyte and neutrophils). Drawing from prior literature [9,18,19,20,21], we also included the following covariates that may explain the association between residential context and epigenetic age: (1) *educational attainment* (no college degree vs. college degree); (2) *homeowner status* (does not own home vs. owns home); (3) *smoker status* (current/former smoker vs. never smoked); and (4) *major life stressors* (additive index). All covariates were measured at Wave 6 except for educational attainment, which ACL measured only once at Wave 1. The index of major life events included being robbed, physically attacked, unexpectedly laid off from a job, or experiencing the death of a parent, child, or friend/other relative since the last interview. Preliminary analyses also included covariates for birth year, family income, marital status, and drinking at Wave 6. These were removed because they did not predict epigenetic age in fully adjusted models.

### 2.3. Analytic Strategies

All statistical analyses were conducted in Stata 18. Our analyses proceeded in two steps. First, we generated descriptive statistics of GrimAge acceleration and covariates, split by respondents’ racial–ethnic and residential segregation statuses. These analyses included means and frequencies of continuous and categorial variables, respectively, along with standard deviations and percentages in parentheses (Table 1). Our goal was to establish descriptive profiles of Black and White respondents who lived in different residential contexts.

Next, we tested a series of linear multilevel regression models of GrimAge acceleration that included random intercepts at the state level (Table 2). Model 1 regressed GrimAge acceleration on dummy variables for sex and race/residential segregation groups, with men and White respondents in segregated White areas as respective reference groups. Models 2 through 5 added educational attainment, homeowner status, smoker status, and traumatic stressors, respectively, in a stepwise fashion. Model 6 added all covariates concurrently. Random intercepts for states were included to account for distinct policy contexts and legacies of segregation across regions. That is, we attempted to account for the possibility that Black residents in highly segregated areas in Deep South states, for example, may have experienced neighborhood and life course contexts distinct from their segregated Black peers in Midwestern or Northeastern states.

Our goal for the second phase of analysis was to determine whether and to what extent residential segregation accounted for Black–White disparities in epigenetic age acceleration, net of state-level clustering and individual covariates. We reported regression coefficients and 95% confidence intervals, which we interpreted as years of accelerated DNA methylation age relative to the reference group. We also report on sensitivity analyses that replicated our models with additional epigenetic clocks (Table 3) and more conservative residential segregation thresholds (Table 4).

## 3. Results

### 3.1. Descriptive Statistics

Table 1 reports descriptive statistics of study variables split by respondents’ racial–ethnic and residential segregation statuses. Across residential contexts, Black respondents typically exhibited around 2 years of GrimAge acceleration, while their White counterparts showed signs of normal or slightly decelerated aging (i.e., scores near or below zero). Regardless of residential context, Black respondents were also less likely than their White peers to have a college degree or own a home. However, compared with all other groups, Black respondents in segregated Black areas were the least likely to own a home (59.3%), most likely to be a former/current smoker (70.4%), and most likely to report multiple (2+) traumatic stressors (37%).

### 3.2. Multilevel Regressions of GrimAge Acceleration

Table 2 reports the results of our multilevel regression models. Adjusting for respondent sex and state-level clustering in Model 1, Black participants in segregated Black areas (b = 3.13; 95% CI = 1.22, 5.04) and non-segregated areas (b = 3.10; 95% CI = 1.20, 4.99) exhibited 3 years of GrimAge acceleration, on average, relative to White participants in segregated White areas. Across Models 2 through 5, having no college degree (vs. college degree), not owning a home (vs. owning a home), being a current/former smoker (vs. never smoking), and reporting 3+ traumatic stressors (vs. none) all predicted significant GrimAge acceleration, at magnitudes ranging from 2.3 to 4.4 years, on average. After including all covariates in Model 6, Black participants living in segregated Black areas no longer exhibited significant GrimAge acceleration, relative to their White peers in segregated White areas (b = 1.30; 95% CI = −0.44, 3.04). However, compared with White participants in segregated White areas, Black respondents in non-segregated areas still expressed 2.05 years of accelerated GrimAge even after accounting for all covariates (b = 2.05; 95% CI = 0.34, 3.76). We found no differences in GrimAge acceleration between White participants across residential contexts.

Finally, supplemental analyses provide additional context and support for our findings. Consistent with prior studies [17,39], we found that the above patterns only surfaced for GrimAge acceleration and did not replicate with the other epigenetic clocks (Table 3). Second, our findings also replicated when we applied a more conservative threshold (Z = 2.58) for the Getis-Ord segregation measure (Table 4).

## 4. Discussion

Black Americans continue to live sicker and shorter lives than their White counterparts. Our study examined residential segregation as an explanation for these patterns. With a focus on biological aging, we tested a series of multilevel models to account for Black–White disparities in epigenetic age acceleration among older-age adults in the Americans’ Changing Lives cohort. Our analyses revealed three key findings.

First, in models adjusting for respondent sex and state-level clustering, we found that older-age Black adults expressed three years of epigenetic age acceleration (i.e., GrimAge), relative to their White counterparts. This baseline disparity persisted regardless of residential segregation. Still, the explanations for this initial disparity appeared to vary depending on residential context.

For instance, our second finding was that Black participants residing in segregated Black areas were much less likely to have a college degree or own a home, and more likely to smoke and report multiple traumatic stressors, relative to their White counterparts in segregated White areas. Moreover, these other conditions ultimately explained the significant gap in epigenetic age acceleration between these two groups.

However, our third finding revealed that Black participants in non-segregated areas expressed patterns distinct from their segregated Black counterparts. Although the former group was also less likely to have a college degree or own a home, they tended to report similar rates of smoking and traumatic stressors as White respondents. Additionally, non-segregated Black respondents still expressed two years of accelerated epigenetic age, relative to segregated White participants, even after accounting for these other factors.

Depressed rates of homeownership among segregated Black respondents likely proxied broader disparities in wealth, rent burdens, and vulnerability to eviction, all of which contribute to early morbidity and mortality among Black populations [19,40]. Tobacco outlets are also overly concentrated in majority-Black areas nationwide [20], and studies indicate that an increase in concentration of local tobacco retailers can induce resident demand in tobacco products [41]. Thus, heightened vulnerability to housing insecurity and other social stressors, coupled with easier access to tobacco, may have formed a toxic cocktail conducive to accelerated aging among segregated Black adults. Future research and policy interventions focused on the health of segregated Black communities should consider examining these conditions in more detail.

Our findings for non-segregated Black adults are consistent with a growing body of research on the population health consequences of structural racism [42]. One takeaway from this literature is that numerous social, cultural, and political–economic forces operate confluently to benefit the health and longevity of White Americans at the expense of other groups [43]. This could explain why non-segregated Black adults also showed signs of accelerated aging. Within affluent and predominantly White contexts, for example, White occupants often enact cultural racism (e.g., anti-Black stigma) to guard the space from outsiders [44,45]. Studies drawing from theories of cultural racism also indicate that Black residents in racially mixed or predominantly White contexts tend to report health complications from chronic racism-related vigilance and stress [46,47,48]. Future research and policy should continue working on residential segregation, while also keeping in mind additional stress and health mechanisms stemming from broader structural racism.

## 5. Limitations

Our study has at least three key limitations. First, Wave 6 of ACL includes a small and highly selective sample of older-age Black and White adults. Indeed, 60% of the original sample has died by Wave 6. Thus, our estimates of epigenetic age disparities are most likely conservative, given that many Black respondents who lived in segregated Black areas earlier in the life course likely died prior to our study period. Our small sample size also limited the number of covariates we could include in our multivariable models.

Second, our data were cross-sectional. Our ability to adjust for residential selection processes and other sources of confounding was thus limited. For example, our findings likely reflect cumulative and bidirectional health and residential mobility processes stemming from earlier life course periods [49,50]. Future work should employ repeated measures of residential contexts and epigenetic age over the life course.

Third, our measurement of segregation is also limited in several respects. For instance, relying on census data to gauge segregation almost invariably introduces measurement error, as measures like these often inaccurately reflect residents’ daily residential experiences and activity spaces [51,52]. Additional research is needed to establish which types of spatial measures are most relevant for testing hypotheses concerning health and aging disparities. Future research is also needed to determine if and how broader sociopolitical contexts, such as state or county policies, condition local patterns of residential segregation and their consequences for resident health [53]. Although our models attempted to account for this possibility using state-level random intercepts, current data limitations precluded us from directly measuring these broader contexts.

## 6. Conclusions

Black–White health disparities persist in the United States. Our findings suggest that residential segregation only partially explains these patterns. We conclude that residential segregation is just one facet of broader racialized social systems undermining the health and longevity of Black Americans in the contemporary United States. Future research and policy should continue probing residential segregation as a social determinant of racialized health disparities, while also keeping in mind that segregation is just one manifestation of broader structural racism.

## Figures and Tables

**Table 1 ijerph-22-00837-t001:** Descriptive Statistics by Respondent Race and Residential Segregation Status: Americans’ Changing Lives, Wave 6 (2019; N = 288).

	Black		White		Total
	High Clustering ^a^	No Clustering ^a^	High Clustering ^a^	No Clustering ^a^	
Chronological Age (years)	69.26 (6.40)	70.63 (8.19)	69.86 (9.83)	71.22 (9.23)	70.68 (9.03)
GrimAge Acceleration (years)	1.54 (5.65)	2.27 (5.19)	−0.19 (5.15)	−0.69 (4.46)	−0.06 (4.90)
College Degree (Wave 1)					
Yes	3 (11.1%)	3 (10.0%)	20 (31.7%)	56 (33.3%)	82 (28.5%)
No	24 (88.9%)	27 (90.0%)	43 (68.3%)	112 (66.7%)	206 (71.5%)
Homeowner Status					
Owns home	16 (59.3%)	20 (66.7%)	54 (85.7%)	141 (83.9%)	231 (80.2%)
Does not own home	11 (40.7%)	10 (33.3%)	9 (14.3%)	27 (16.1%)	57 (19.8%)
Smoker Status					
Never smoked	8 (29.6%)	13 (43.3%)	31 (49.2%)	77 (45.8%)	129 (44.8%)
Current/former smoker	19 (70.4%)	17 (56.7%)	32 (50.8%)	91 (54.2%)	159 (55.2%)
Major Life Stressors					
0	6 (22.2%)	14 (46.7%)	16 (25.4%)	55 (32.7%)	91 (31.6%)
1	11 (40.7%)	12 (40.0%)	38 (60.3%)	70 (41.7%)	131 (45.5%)
2	9 (33.3%)	4 (13.3%)	9 (14.3%)	36 (21.4%)	58 (20.1%)
3+	1 (3.7%)	0 (0.0%)	0 (0.0%)	7 (4.2%)	8 (2.8%)
Respondent Sex					
Male	10 (37.0%)	15 (50.0%)	32 (50.8%)	81 (48.2%)	138 (47.9%)
Female	17 (63.0%)	15 (50.0%)	31 (49.2%)	87 (51.8%)	150 (52.1%)
**Total**	27 (9.4%)	30 (10.4%)	63 (21.9%)	168 (58.3%)	288 (100.0%)

Notes: Means/frequencies reported with standard deviations/percentages in parentheses. All measures are recorded at Wave 6 unless stated otherwise. ^a^ Black and White participants who reside (vs. do not reside) in highly clustered Black and White census tracts, respectively, based on a Getis-Ord z-statistic greater than 1.96.

**Table 2 ijerph-22-00837-t002:** Nested Multilevel Models of GrimAge Acceleration: Americans’ Changing Lives, Wave 6 (2019; N = 288).

	Model 1		Model 2		Model 3		Model 4		Model 5		Model 6	
Respondent Race/Residential Segregation (ref. = White/high clustering) ^a^												
Black/high clustering	3.131	**	2.720	**	2.349	*	2.366	**	2.879	**	1.301	
	[1.219, 5.044]		[0.844, 4.595]		[0.474, 4.224]		[0.580, 4.152]		[0.969, 4.788]		[−0.435, 3.038]	
Black/no clustering	3.095	**	2.591	**	2.487	**	2.872	**	3.088	**	2.051	*
	[1.198, 4.992]		[0.739, 4.442]		[0.625, 4.349]		[1.109, 4.636]		[1.202, 4.974]		[0.344, 3.757]	
White/no clustering	−0.638		−0.593		−0.620		−0.724		−0.864		−0.888	
	[−1.854, 0.579]		[−1.772, 0.587]		[−1.799, 0.559]		[−1.853, 0.405]		[−2.079, 0.351]		[−1.968, 0.193]	
No college degree (vs. college degree)	--		2.274	**	--		--		--		1.355	**
			[1.172, 3.377]								[0.342, 2.368]	
Does not own home (vs. owns home)	--		--		2.749	**	--		--		2.329	**
					[1.551, 3.947]						[1.234, 3.424]	
Current/former smoker (vs. never smoked)	--		--		--		3.186	**	--		2.816	**
							[2.274, 4.098]				[1.940, 3.692]	
Major life stressors (ref. = none)												
1	--		--		--				−0.096		−0.187	
									[−1.204, 1.013]		[−1.172, 0.798]	
2	--		--		--				0.290		0.283	
									[−1.087, 1.667]		[−0.945, 1.511]	
3+	--		--		--				4.404	**	3.572	**
									[1.409, 7.399]		[0.906, 6.238]	
Female (vs. male)	−3.028	**	−3.630	**	−3.228	**	−2.812	**	−2.872	**	−3.237	**
	[−4.008, −2.049]		[−4.629, −2.632]		[−4.178, −2.278]		[−3.722, −1.901]		[−3.849, −1.894]		[−4.151, −2.323]	
Intercept	1.246	*	−0.002		0.937		−0.471		1.174		−1.272	
	[0.069, 2.423]		[−1.276, 1.272]		[−0.230, 2.104]		[−1.674, 0.731]		[−0.242, 2.590]		[−2.697, 0.152]	

Notes: Unstandardized linear coefficients are reported with 95% confidence intervals in brackets. All models adjust for cell composition and random intercepts at the state level (n = 39). ^a^ Black and White participants who reside (vs. do not reside) in highly clustered Black and White census tracts, respectively, based on a Getis-Ord z-statistic greater than 1.96. ** *p* < 0.01, * *p* < 0.05 (two-tailed).

**Table 3 ijerph-22-00837-t003:** Multilevel Models of Additional Epigenetic Age Clocks: Americans’ Changing Lives, Wave 6 (2019; N = 288).

	PhenoAge Acceleration	SkinBloodAge Acceleration	HannumAge Acceleration		HorvathAge Acceleration
Respondent Race/Residential Segregation ^a^ (ref. = White/high clustering)					
Black/high clustering	0.532	−0.077	−1.976		−0.994
	[−2.196, 3.260]	[−1.755, 1.600]	[−3.955, 0.004]		[−3.070, 1.083]
Black/no clustering	1.180	−0.328	−1.543		0.690
	[−1.480, 3.839]	[−1.964, 1.308]	[−3.473, 0.387]		[−1.351, 2.732]
White/no clustering	−1.022	−0.278	−0.898		0.376
	[−2.740, 0.697]	[−1.335, 0.779]	[−2.145, 0.349]		[−0.938, 1.690]
Intercept	0.508	0.617	1.365	*	0.239
	[−1.108, 2.123]	[−0.376, 1.611]	[0.193, 2.537]		[−1.012, 1.490]

Notes: Unstandardized linear coefficients are reported with 95% confidence intervals in brackets. All models adjust for respondent sex, cell composition, and random intercepts at the state level (n = 39). ^a^ Black and White participants who reside (vs. do not reside) in highly clustered Black and White census tracts, respectively, based on a Getis-Ord z-statistic greater than 1.96. * *p* < 0.05 (two-tailed).

**Table 4 ijerph-22-00837-t004:** Alternative Models of GrimAge Acceleration Using a Conservative Getis-Ord Segregation Threshold: Americans’ Changing Lives, Wave 6 (2019; N = 288).

	Model 1		Model 2		Model 3		Model 4		Model 5		Model 6	
Respondent Race/Residential Segregation ^a^ (ref. = White/high clustering)												
Black/high clustering	2.669	*	2.224	*	1.958		2.215	*	2.534	*	1.248	
	[0.414, 4.923]		[0.017, 4.431]		[−0.241, 4.157]		[0.118, 4.312]		[0.285, 4.783]		[−0.784, 3.280]	
Black/no clustering	3.108	**	2.674	**	2.583	**	2.869	**	2.948	**	2.038	*
	[1.125, 5.092]		[0.745, 4.603]		[0.641, 4.525]		[1.022, 4.716]		[0.985, 4.910]		[0.267, 3.810]	
White/no clustering	−0.769		−0.700		−0.645		−0.653		−1.001		−0.730	
	[−2.265, 0.727]		[−2.148, 0.748]		[−2.099, 0.809]		[−2.044, 0.739]		[−2.486, 0.484]		[−2.055, 0.596]	
Intercept	1.395		0.126		0.999		−0.454		1.309		−1.341	
	[−0.034, 2.823]		[−1.370, 1.622]		[−0.415, 2.413]		[−1.890, 0.982]		[−0.279, 2.898]		[−2.920, 0.238]	

Notes: Unstandardized linear coefficients are reported with 95% confidence intervals in brackets. All models adjust for cell composition, covariates, and random intercepts at the state level (n = 39). ^a^ Black and White participants who reside (vs. do not reside) in highly clustered Black and White census tracts, respectively, based on a Getis-Ord z-statistic greater than 2.58. ** *p* < 0.01, * *p* < 0.05 (two-tailed).

## Data Availability

Deidentified survey data from the Americans’ Changing Lives (ACL) study are publicly accessible via the Inter-university Consortium for Political and Social Research (ICPSR; https://www.icpsr.umich.edu, accessed on 22 May 2025). Deidentified geospatial data from the ACL study are publicly accessible via the virtual data enclave of the ICPSR (https://www.icpsr.umich.edu/web/pages/ICPSR/access/restricted/enclave.html, accessed on 22 May 2025). Deidentified DNA methylation data from the ACL will be made publicly accessible via the National Institutes of Health dbGAP database (https://www.ncbi.nlm.nih.gov/gap/, accessed on 22 May 2025).

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
