# Peer review of "Residential Segregation and Epigenetic Age Acceleration Among Older-Age Black and White Americans"

_ijerph, 2025, doi:10.3390/ijerph22060837_

Round 1

Reviewer 1 Report

Comments and Suggestions for Authors

All comments intended for the authors are provided in the attached text.

Author Response

[A1]: The manuscript would benefit from a brief explanation of the spatial extent of the research. At least one sentence clarifying the geographic scope of the study area would enhance the reader's understanding of the context.

Thank you for spotting this. We added the following information (underlined) to the abstract: "We analyze data from 288 Black and White older-age adults who participated in Wave 6 (2019) of the Americans’ Changing Lives study, a nationally representative sample of adults in the contiguous United States."

[A2]: This statement requires citation. Instead, I would suggest beginning the manuscript — particularly the abstract — with a more general statement about the aging process among Black Americans. Accordingly, it is also necessary to revise the second sentence of the abstract, where you could introduce the disparity between Black and White Americans.

We removed this sentence from the abstract. The first sentence of the abstract now reads: "Our study tests residential segregation as an explanation for biological aging disparities between Black and White Americans."

[A3]: Since these terms are being mentioned for the first time in the text, please provide their full names, followed by the abbreviations in parentheses.

We revised this sentence as follows: "Our outcome of interest is epigenetic age acceleration assessed via five epigenetic clocks: GrimAge, PhenoAge, SkinBloodAge, HannumAge, and HorvathAge."

We now specify that our outcomes are "five epigenetic clocks." The names we use to refer to these clocks are the names used in the literature.

[A4]: I would recommend adding an additional sentence in the abstract to briefly highlight the implications of your research.

Thank you for this suggestion. We added the following sentence to the end of the abstract: "Our findings suggest residential segregation only partially explains why Black Americans tend to live shorter lives than White Americans."

[A5]: The introduction lacks references to prior studies that have addressed this topic. Including a brief review of the
relevant literature — ideally in a separate “Literature Review” section — would strengthen the manuscript. I particularly
recommend citing more recent sources, preferably within the last five years. Additionally, the introduction does not clearly articulate the research gap that this study aims to address. Once the authors present previous findings, they can then elaborate on what those studies have lacked and how their own work addresses those gaps. I believe this is a crucial element for both the introduction and the subsequent discussion section. 

Thank you for catching this critical omission. We have added the following text (underlined) to our introduction:

"Our study advances the above aims by being one of few to test residential segregation as a social determinant of epigenetic aging disparities among a national sample of older-age Black and White adults 13,15. Indeed, a scoping review found only nine studies conducted on the topic of neighborhood determinants of epigenetic age since early 2022.16 But none of these studies focused specifically on residential segregation. Following the 2022 review, one recent study found that older-age Black residents who lived in segregated and impoverished Black neighborhoods exhibited signs of accelerated epigenetic age, relative to their Black peers in more advantaged neighborhoods. 17 However, this study only included six US cities, whereas our analysis represents the entire contiguous US."

[A6]: I kindly ask the authors to provide a source for this statement.

Thank you for noticing this omission. A reference has been added to this statement.

[A7]: Could you briefly explain why other racial-ethnic groups were excluded from the analyses.

Thank you for this suggestion. We now include the following rationale for excluding other groups: "Other racial-ethnic groups were excluded from our analyses due to small cell sizes, and because our focus is Black-White disparities."

[A8]: Could you please add the source here? Which researchers are you referring to here?

Thank you for identifying this omission. We added a reference to Kershaw et al. 2017.

[A9]: Please, add the source for the program.

Thank you for catching this. We added a reference to Moran et al. 2016.

[A10]: Perhaps you could consider linking this sentence to stress, in light of the events mentioned.

Thank you for catching this inconsistency. We renamed this variable "major life stressors."

[A11]: The Results section appears somewhat brief in comparison to the preceding Materials and Methods section.

The first version of the paper included the wrong tables in the manuscript. We fixed this error so that our discussion of main findings aligns with the correct tables.

[A12]: Would it be possible for the authors to include a map that clearly shows the differences between Black and White Americans, based on the results of the Getis-Ord statistic? This would enrich the Results section, as the map could present the findings in a simple yet sophisticated way.

We agree with the reviewer that providing a visual representation of our findings would be ideal. However, our measure of residential segregation is recorded at the census tract level. Unfortunately, it would not be feasible to provide an intelligible US map disaggregated by census tract boundaries.

[A13]: Since you used the term 'studies', please include at least one more reference. Also, this sentence is more appropriate for the Discussion section, as you are making a comparison with other studies.

Thank you for catching this. We added a second citation (Hicken et al. 2023).

[A14]: It would be helpful to number the other two key findings as well (second; third) to improve clarity and readability.

The revised manuscript now numbers each of the three key findings.

[A15]: Could you kindly rephrase the sentence?

We changed this sentence as follows: "Second, we found that Black participants residing in segregated Black areas were much less likely to have a college degree or own a home, and more likely to smoke and report multiple traumatic stressors, relative to their White counterparts in segregated White areas."

[A16]: This part should be further elaborated—namely, the comparison of the findings with those of other studies. It would be useful to indicate which studies show similar results and which ones differ from this research.

We added two additional citations to this sentence. We also note that we included the following statements about similarities with prior research in this area:

"Within affluent and predominantly White contexts, for example, White occupants often enact cultural racism (e.g., anti-Black stigma) to guard the space from outsiders.41,42 Studies drawing from theories of cultural racism also indicate that Black residents in racially mixed or predominantly White contexts tend to report health complications from chronic racism-related vigilance and stress.43–45"

We are suggesting that these unmeasured processes could explain why Black respondents living in non-segregated areas also exhibit accelerated epigenetic age relative to White counterparts.

[A17]: Please consider to move this sentence to the Limitations section, at the end, where you refer to future research.

Thank you for this suggestion. We decided to keep this sentence here because it is directly related to research on structural racism.

[A18]: Please, change the year – 1992.

Thank you for spotting this error. We changed the year to 1992.

Reviewer 2 Report

Comments and Suggestions for Authors

Some methodological approaches raise questions. Firstly, «(3) smoker status (current/former smoker vs. never smoked);» The study by Joehanes R, et al (2016) shows that «Methylation levels of most CpGs returned toward that of never-smokers within five years of smoking cessation». Therefore, it would be more correct to take into account the duration of the smoking cessation period in the analysis. Secondly, «for educational attainment, which ACL measured only once at Wave 1.» «The first wave of data collection (1986) recruited adults aged 25 and older» That is, it is assumed that at the age of 25 and older no one has received higher education? How correct is this assumption? Another problem is the small size of the group, which negatively affects the results of the analysis. For example, «Preliminary analyses also included covariates for … drinking... These were removed because they did not predict epigenetic age in final models.» Although it is known that alcohol dependence is associated with accelerated aging. Shirai T, et al (2024): "We observed that DunedinPACE accelerated more in patients with alcohol dependence. AgeAccelGrim and AgeAccelGrim2 decelerated more after the treatment program than before". The insufficient size of the study group did not allow the authors to detect an association of accelerated aging with alcohol consumption. Accordingly, other factors could have been missed. Therefore, on the one hand, the study is of some interest, on the other hand, it has serious limitations. But perhaps, having been published, it will someday be included in the next meta-analysis and thus bring some benefit.

Joehanes R, et al  Epigenetic Signatures of Cigarette Smoking. Circ Cardiovasc Genet. 2016 Oct;9(5):436-447. doi: 10.1161/CIRCGENETICS.116.001506. Epub 2016 Sep 20. PMID: 27651444; PMCID: PMC5267325.

Shirai T, Okazaki S, Otsuka I, Miyachi M, Tanifuji T, Shindo R, Okada S, Minami H, Horai T, Mouri K, Hishimoto A. Accelerated epigenetic aging in alcohol dependence. J Psychiatr Res. 2024 May;173:175-182. doi: 10.1016/j.jpsychires.2024.03.025. Epub 2024 Mar 23. PMID: 38547739.

Author Response

Response 1: Some methodological approaches raise questions. Firstly, «(3) smoker status (current/former smoker vs. never smoked);» The study by Joehanes R, et al (2016) shows that «Methylation levels of most CpGs returned toward that of never-smokers within five years of smoking cessation». Therefore, it would be more correct to take into account the duration of the smoking cessation period in the analysis. Secondly, «for educational attainment, which ACL measured only once at Wave 1.» «The first wave of data collection (1986) recruited adults aged 25 and older» That is, it is assumed that at the age of 25 and older no one has received higher education? How correct is this assumption? Another problem is the small size of the group, which negatively affects the results of the analysis. For example, «Preliminary analyses also included covariates for … drinking... These were removed because they did not predict epigenetic age in final models.» Although it is known that alcohol dependence is associated with accelerated aging...The insufficient size of the study group did not allow the authors to detect an association of accelerated aging with alcohol consumption. Accordingly, other factors could have been missed. Therefore, on the one hand, the study is of some interest, on the other hand, it has serious limitations. But perhaps, having been published, it will someday be included in the next meta-analysis and thus bring some benefit.

We thank the reviewer for raising these valid concerns. Unfortunately, the ACL study does not ask respondents how long they have been smoking, only whether they are a current or former smoker. Likewise, ACL measured educational attainment only once at Wave 1. We also agree with the reviewer that the small sample size is a major limitation of our study. We decided to add the following text (underlined) to our discussion of limitations:

"Our study has at least three key limitations. First, Wave 6 of ACL includes a small and highly selective sample of older-age Black and White adults. Indeed, 60% of the original sample has died by Wave 6. Thus, our estimates of epigenetic age disparities are most likely conservative, given that many Black respondents who lived in segregated Black areas earlier in the life course likely died prior to our study period. Our small sample size also limited the number of covariates we could include in our multivariable models."